# A Novel NIR Fluorescent Probe for Highly Selective Detection of Nitroreductase and Hypoxic-Tumor-Cell Imaging

**DOI:** 10.3390/molecules26154425

**Published:** 2021-07-22

**Authors:** Feng Liu, Hong Zhang, Kun Li, Yongmei Xie, Zhihui Li

**Affiliations:** 1Department of Thyroid Surgery, West China Hospital of Sichuan University, Chengdu 610041, China; liufenghxjr@163.com (F.L.); xieym@scu.edu.cn (Y.X.); 2Laboratory of Thyroid and Parathyroid Disease, Frontiers Science Center for Disease-Related Molecular Network, West China Hospital of Sichuan University, Chengdu 610041, China; 3College of Chemistry, Sichuan University, Chengdu 610041, China; 2019322030029@stu.scu.edu.cn (H.Z.); kli@scu.edu.cn (K.L.)

**Keywords:** nitroreductase, NIR, hypoxic, cell imaging

## Abstract

Nitroreductase as a potential biomarker for aggressive tumors has received extensive attention. In this work, a novel NIR fluorescent probe for nitroreductase detection was synthesized. The probe **Py-SiRh-NTR** displayed excellent sensitivity and selectivity. Most importantly, the confocal fluorescence imaging demonstrated that HepG-2 cells treated with **Py-SiRh-NTR** under hypoxic conditions showed obvious enhanced fluorescence, which means that the NTR was overexpressed under hypoxic conditions. Moreover, the probe showed great promise that could help us to study related anticancer mechanisms research.

## 1. Introduction

Nitroreductase (NTR), as a family of evolution-related proteins, is involved in the reduction of nitrogen-containing compounds [1,2]. NTR has received extensive attention as a potential biomarker for aggressive tumors in recent years [3]. Moreover, NTR can be overexpressed in hypoxic tumors [4]. Hypoxia is a feature of tumor tissue, whereby the median oxygen (O_2_) concentration in some solid tumors is ∼4% and may even decrease to 0% locally [5,6,7]. In clinical applications, the formation of hypoxic tumors with abnormal microvessels can limit the perfusion of cytotoxic chemotherapeutic drugs, therein decreasing their curative effects compared with that of normoxic tumors [5,6,7]. Therefore, estimating the tumor hypoxia degree is of great importance in predicting anticancer efficacies [7]. There is no doubt that developing an effective method to monitor NTR activity is not only useful for clinical diagnoses and anticancer mechanisms research, but also for new anticancer drug evaluation and tumor therapy.

Various methods have been developed for the detection and identification of NTR. The fluorescent imaging has attracted much attention as a powerful technique in monitoring biological analytes and processes in living systems [8]. Especially, the near-infrared (NIR) bioprobes are advantageous because of the obvious advantages of NIR light including minimum photodamage to biological samples, a high signal-to-noise ratio, and deep tissue penetration [9,10]. The detection of NTR activities within hypoxic tumor cells and tissues using fluorescent methods has been well described recently [11]. However, most of these NTR fluorescent probes need ultraviolet light as the excitation light which could cause photodamage to biological samples. Most importantly, only a few fluorescent probes have the ability to target the activity of NTR in specific subcellular organelle [12,13,14,15,16,17,18,19,20,21,22,23,24,25].

Herein, in this work, we constructed such a novel NIR fluorescent probe to detect nitroreductase (**Py-SiRh-NTR**; Scheme 1). Our design strategy could be finished via three-step synthesis to obtain the benzyl pyridinium salt to construct **Py-SiRh-NTR**. Further, we hypothesized that the benzyl pyridinium salt group could be triggered by nitroreductase, followed by the linker self-immolation to form **Py-SiRh** with strong near-infrared fluorescence emission.

## 2. Results and Discussion

Firstly, we tested the response of **Py-SiRh-NTR** to NTR. The probe was incubated in 10 mM PBS buffer solution at 37 °C, containing 200 µM NADH. The fluorescence spectra were depicted in Figure 1a; the fluorescent probe **Py-SiRh-NTR** showed a very weak fluorescence at 680 nm. After the addition of NTR, the solution showed an obvious fluorescence enhancement, and the fluorescence intensity reached the maximum after 180 min. The titration experiments indicated that a higher NTR concentration induced the significant fluorescence enhancement (28-fold). Especially, the fluorescence intensity has an excellent linear relationship with the concentrations of NTR in 0−10 μg/mL (Figure 1b). The detection limit was calculated to be as low as 0.07 μg/mL. The results indicated that the sensitivity of **Py-SiRh-NTR** is comparable to the published NTR probes. Moreover, the solution of **Py-SiRh-NTR** has no absorption at about 655 nm, but after the addition of NTR, the absorption at 655 nm increased distinctly (Figure 1c). The reaction product with NTR was verified by high resolution mass spectrometry. A peak at *m*/*z* = 386.2045 assigned to **Py-SiRh** was observed, that verified the proposed response mechanism. Thus, we could assuredly say that the proposed response mechanism was well demonstrated.

Then, the time-dependent fluorescence response of **Py-SiRh-NTR** was examined (Appendix A). All of the fluorescence kinetic curves are shown in Figure 1d. It was apparent to see that the fluorescence intensity of **Py-SiRh-NTR** depended on the concentration of NTR. In other words, a higher concentration of NTR could induce the faster fluorescence intensity, enhanced. The time taken to reach the reaction equilibrium of **Py-SiRh-NTR** was tested for 180 min. Furthermore, the kinetic parameters for this NTR-catalyzed reaction were also verified. A Lineweaver–Burke plot of 1/V (V is the initial reaction rate) versus the reciprocal of the probe **Py-SiRh-NTR** concentration is shown in Appendix A [26]. According to the Michaelis–Menten equation [26,27], the Michaelis constant (Km) and maximum of the initial reaction rate (Vmax) were examined to be 52.2 μM and 0.074 μM/S, respectively. 

Subsequently, the selectivity of the probe was tested for NTR over other biologically relevant species, such as trypsin, lipase, bull serum albumin (BSA), pepsin, amino acids (cysteine, homocysteine, glutathione) and some inorganic salts (K^+^, Na^+^, Fe^2+^). The results were shown in Figure 2, and it was apparent to see that only the NTR could make an obvious fluorescence enhancement compared to the controls. It was demonstrated that the ***Py-SiRh-NTR*** is highly selective for NTR over other typical physiological species, which may be necessary for the bio-imaging in tangled living cells. All of the results of the experiments clearly demonstrated that this NIR fluorescent probe can be an excellent tool to detect the NTR in living cells.

Encouraged by the fluorescent experiments, we attempted to examine whether the **Py-SiRh-NTR** could perform well in living cells. Before the bioimaging, a standard MTS assay was carried out to test the biocompatibility. The results revealed that the viabilities of HepG-2 cells were not noticeably affected by incubation with different concentrations of **Py-SiRh-NTR** (1.0–10.0 μM) for 24 h (Figure 3). Thus, the low cytotoxicity of the probe in the measured concentration range is well demonstrated. Therefore, in the subsequent bio-imaging experiments, 2 μM **Py-SiRh-NTR** was thus used.

Then, we used the confocal fluorescence microscopy on HepG-2 cells to test the imaging ability of **Py-SiRh-NTR** for the intracellular NTR monitoring, as we know that the tumor cells could overexpress NTR under hypoxic conditions. Therefore, in this experiment, the HepG-2 cells were incubated with 2 μM **Py-SiRh-NTR** and treated under normoxic conditions (20% O_2_) and hypoxic conditions (1% O_2_) for 4 h to compare the concentration of NTR under different oxygen conditions. The results are shown in Figure 4; HepG-2 cells incubated under normoxic conditions showed a very weak near-infrared fluorescence that indicated the low concentration of NTR. On the contrary, HepG-2 cells displayed increased near-infrared fluorescence intensity when treated under hypoxic conditions, indicating that the HepG-2 cells could overexpress NTR in hypoxic conditions. Most importantly, when the HepG-2 cells were pretreated with the dicoumarin, a well-known NTR inhibitor under hypoxic conditions for 4 h [28], no obvious near-infrared fluorescence signal was detected. This verified that the near-infrared fluorescence signal was exactly from the production of the **Py-SiRh-NTR** with NTR in HepG-2 cells under hypoxic conditions. The results showed that the **Py-SiRh-NTR** could monitor the activity of nitroreductase in living cells and evaluate the degree of tumor hypoxia. 

We also examined the lysosome-targeting ability of the probe after reaction with NTR. HepG-2 cells were treated with **Py-SiRh-NTR** under the hypoxic condition of 1% O_2_ for 4 h, according to our confocal fluorescence microscopy results, and then incubated with LysoTracker green (0.5 μM, 15 min). As shown in Figure 5, the excellent overlapping images of the NIR fluorescence channel and green fluorescence channel displayed a high Pearson coefficient, which indicated that **Py-SiRh-NTR** could specifically localize in lysosomes after the probe finished the reaction with the NTR under the hypoxic condition. Finally, we also tested the co-localization ability with KTC-1, C643 and B-CPAP cells, and achieved a good Pearson coefficient (Figure 6). 

## 3. Conclusions

In summary, we have designed the **Py-SiRh-NTR** as a brand new NIR fluorescent probe for highly selective detection of nitroreductase and hypoxic-tumor-cell imaging. The probe shows good solubility and is highly sensitive to NTR, which could be used to discern the degree of hypoxia tumor cells. To be specific, the **Py-SiRh-NTR** provides an outstanding fluorescent diagnostic tool which can be used to study the NTR activity-related pathological process of cancers. The bright potential diagnostic application of **Py-SiRh-NTR** is apparent.

## Data Availability

Data present in this study are available on request from the corresponding author.

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
