# Peer review of "A Novel NIR Fluorescent Probe for Highly Selective Detection of Nitroreductase and Hypoxic-Tumor-Cell Imaging"

_molecules, 2021, doi:10.3390/molecules26154425_

Round 1

Reviewer 1 Report

This paper describes the design, the synthesis and the evaluation of a fluorescent nitroreductase (NTR) probe called Py-SiRh-NTR. This probe bears a nitro group which could be reduced by NTR. After subsequent linker self-immolation, the probe led to Py-SiRh which is a strong NIR emissive probe. The design, the synthesis and the in vitro evaluation of the probe is well described. Moreover, the probe has been imaged under normal versus hypoxia conditions in living cells. This demonstrated that the probe is able to detect NTR in cells and as NTR is overexpress in hypoxia conditions, this demonstrated also that the probe is able to detect hypoxia.

Even though a lot of experimental work and different techniques have been used, the main message of the paper is not clear and the title should be modified. The issue is that there is no evidence that the probe Py-SiRh-NTR is localized in lysosomes. Moreover, Py-SiRh-NTR possess a pyridinium moiety which is known to target towards mitochondria and this probe does not possess any lysosome targeting moiety. Due to its pyridinium moiety, this probe could be expected to localize in mitochondria. The probe could then be reduced by mitochondrial NTR to release Py-SiRh. The fluorescent released probe Py-SiRh can then reach lysosomes as it has been published that Py-SiRh targets lysosomes (Chinese Chem Lett 2019, 1063). It is quite hard to demonstrate that Py-SiRh-NTR targets lysosomes. To my opinion, this probe is able to detect hypoxic conditions in cells due to its reduction via NTR. I don’t think that we could assess that this reduction occurs in the lysosome.

I have also some more detailed remarks.

  • The references have to be carefully checked. As examples, reference 3 is not focused on hypoxic tumors; reference 11 is not focused on NTR. Reference 10 should be cited on page 2 line 49. A reference should be added for dicoumarin described as a well-known NTR inhibitor. Are references 26 and 27 well cited ?
  • The absorption and emission spectra of already described Py-SiRh should be reminded at least in the Supporting Information.

Author Response

Reviewer 1

Q1. The issue is that there is no evidence that the probe Py-SiRh-NTR is localized in lysosomes. Moreover, Py-SiRh-NTR possess a pyridinium moiety which is known to target towards mitochondria and this probe does not possess any lysosome targeting moiety. Due to its pyridinium moiety, this probe could be expected to localize in mitochondria. The probe could then be reduced by mitochondrial NTR to release Py-SiRh. The fluorescent released probe Py-SiRh can then reach lysosomes as it has been published that Py-SiRh targets lysosomes (Chinese Chem Lett 2019, 1063). It is quite hard to demonstrate that Py-SiRh-NTR targets lysosomes. To my opinion, this probe is able to detect hypoxic conditions in cells due to its reduction via NTR. I don’t think that we could assess that this reduction occurs in the lysosome.

Response: We deeply appreciate the reviewer's comments and constructive suggestions. Actually, there is really no direct evidence that the probe Py-SiRh-NTR is localized in lysosomes, since it doesn’t have fluorescence before it was reduced by nitroreductase. We only can see the high pearson coefficient when Py-SiRh-NTR was transformed to Py-SiRh, which we have proved its lysosome targeting ability in our previous reports (Chinese Chem Lett 2019, 1063). Following the reviewer’s suggestion, we have modified the title as “A Novel NIR Fluorescent Probe for Highly Selective Detection of Nitroreductase and Hypoxic-Tumor-Cell Imaging”. And the related description in the manuscript was also modified.

Q2. The references have to be carefully checked. As examples, reference 3 is not focused on hypoxic tumors; reference 11 is not focused on NTR. Reference 10 should be cited on page 2 line 49. A reference should be added for dicoumarin described as a well-known NTR inhibitor. Are references 26 and 27 well cited?

Response: We deeply appreciate the reviewer's comments and constructive suggestions. The reference 3 was changed as “Chem. Soc. Rev., 2019, 48, 683—722”. The reference 11 was changed as “Y. Liu, L. Teng, L. Chen, H. Ma, H. W. Liu and X. B. Zhang, Chem. Sci., 2018, 9, 5347–5353”. The reference about the NTR inhibitor was add as References 28 “ Chevalier, A., Zhang, Y., Khdour, O.M., Kaye, J.B., Hecht, S.M., J. Am. Chem. Soc. 2016, 138, 12009–12018”

Q3. The absorption and emission spectra of already described Py-SiRh should be reminded at least in the Supporting Information.

Response: Following the reviewer’s suggestion, the absorption and emission spectra of Py-SiRh were added as Figure S3 in the supporting information.

Reviewer 2 Report

In the manuscript titled “A Novel Lysosome-Targeting NIR Fluorescent Probe for Highly 2 Selective Detection of Nitroreductase and Hypoxic-Tumor-Cell Imaging”, by  Zhihui Li and co-workers report a novel lysosome-targeting NIR fluorescent probe for nitroreductase detection. The developed probe Py-SiRh-NTR reportedly demonstrates excellent sensitivity and selectivity and also applied in confocal fluorescence imaging experiments to measure NTR expression under hypoxic conditions. Moreover, the lysosomal localization of the developed probe was also confirmed.

While this study involves the development of a promising NTR fluorescent probe. The study requires further minor experiments to facilitate thorough understanding of probe. Therefore, the manuscript is recommended for publication in Molecules, MDPI after the following minor revisions.

  • For experiments reported in Fig 2, there are no error bars. It is require that the authors perform sufficient replicates against various biologically relevant species to confirm the specificity in a statistically significant manner.
  • The Cytotoxicity of Py-SiRh-NTR in cultured HepG-2 cells reported in Figure 3 is performed at a 24 h time point, further time points also need to be tested particularly the standard 72 h time point.
  • It is recommended that the authors perform the hypoxia conditions imaging at multiple time points to investigate whether the probe is capable of detecting early-stage NTR expression and what the time lines of that expression really are and whether they correspond with te data available in the literature.
  • NTR detection using the developed probe could also be investigated using flow cytometry experiments to further look at a larger population of cells and cell types. Again, flow cytometry epxeriments for hypoxia conditions maintained for different time points might reveal separate populations cells (one that have significant NTR expression and ones that perhaps don’t). These studies will improve the impact of a valuable sensor such as the Py-SiRh-NTR.
  • There are multiple grammatical errors in the manuscript that need to be looked into.
  • The information provided in the supplementary file is very poor. The supplementary information has no details about the cells cultured, culture conditions, experimental details of the imaging systems, imaging conditions, including lasers, detection wavelengths, doses and duration. This information must be included along with any instrumentation and experimental details for the spectroscopy-based experiments reported in all the figures.

Round 2

Reviewer 1 Report

The requested revision have been mainly done. I would suggest to delete the following sentences (just before the Conclusions) : In other words, the Py- SiRh-NTR could greatly facilitate the research of lysosomes-related NTR pathological effects. There is no evidence that the probe goes to the lysosomes before reaction with NTR. This sentence can be misleading.

Some English changes are required: We also examined the lysosome-targeting ability of Py-SiRh-NTR.the probe after reacted with NTR. after reaction 

Co-location should be changed to co-localization.

Author Response

Q1. The requested revision have been mainly done. I would suggest to delete the following sentences (just before the Conclusions) : In other words, the Py- SiRh-NTR could greatly facilitate the research of lysosomes-related NTR pathological effects. There is no evidence that the probe goes to the lysosomes before reaction with NTR. This sentence can be misleading.

Response: We deeply appreciate the reviewer's suggestions. Following the reviewer’s suggestion, we deleted the sentence “In other words, the Py- SiRh-NTR could greatly facilitate the research of lysosomes-related NTR pathological effects.”

Q2. Some English changes are required: We also examined the lysosome-targeting ability of Py-SiRh-NTR.the probe after reacted with NTR. after reaction. Co-location should be changed to co-localization.

Response: Thanks for the reviewer’s careful work. We have checked the manuscript carefully and corrected the mistakes.